



# Block-structured, equal workload, multigrid nesting interface for Boussinesq wave model FUNWAVE-TVD

Young-Kwang Choi[1,2], Fengyan Shi[1], Matt Malej[3], Jane M. Smith[3], James T. Kirby[1], and Stephan T. Grilli[4]

[1]Center for Applied Coastal Research, University of Delaware

[2]Task Force for Construction of RV ISABU Support Facility, Korea Institute of Ocean Science and

Technology, Busan Metropolitan City, Republic of Korea

[3]U.S. Army Engineer Research and Development Center, Coastal and Hydraulics Laboratory, 3909 Halls

Ferry Road, Vicksburg, MS 39180, USA

[4]Department of Ocean Engineering, University of Rhode Island, Narragansett, RI 20882, USA

Corresponding author: Fengyan Shi, `fyshi@udel.edu`



## Abstract

We describe the development of a block-structured, equal CPU-load, multigrid nesting interface for the Boussinesq wave model FUNWAVE-TVD. The new model framework does not interfere with the core solver, and thus the core program, FUNWAVE-TVD, is still a stand-alone model used for a single grid. The nesting interface manages the time sequencing and two-way nesting processes between the parent grid and child grid with grid refinement in a hierarchical manner. Workload balance in the MPI-based parallelization is handled by an equal-load scheme. A strategy of shared array allocation is applied for data management, that allows a large number of nested grids without creating additional memory allocations. Four model tests are conducted to verify the nesting algorithm, model accuracy, wetting-drying treatment, and the robustness in the application to modeling transoceanic tsunamis and coastal effects.

## Plain Language Summary

The multiple-grid nesting technique is an important methodology used for modeling transoceanic tsunamis and coastal effects. The traditional grid nesting approach is one-way nesting, which is done manually grid by grid. In this study, we developed a two-way nesting interface in a multigrid nesting system for the Boussinesq wave model, FUNWAVE-TVD. FUNWAVE-TVD is a widely accepted open-source wave model for simulating surface wave propagation and wave-driven processes in the nearshore region, as well as tsunami wave propagation and evolution from the oceanic scale to nearshore scales. The new development of the interface does not alter the core solver, and thus the core program is still a stand-alone model used for single grid applications. Some strategies in workload balance, data management, and parent-child communications in the MPI-based parallelization system are utilized to guarantee the model efficiency and accuracy. Four model tests are carried out in the paper.

## 1 Introduction

To improve the resilience of the world's highly populated coastal areas to tsunami hazard when tsunamigenic events (typically earthquakes or landslides) occur, there has been an increasing need for issuing early warnings and near- and far-field forecast of tsunami coastal impact. This has led to a growing demand for accurate and efficient models of transoceanic tsunami propagation, in multiple-nested grid systems that allow refining the



discretization towards shore, as depth decreases. Models predicting tsunami wave evolution from generation at the source, to propagation at the oceanic basin-scale, transformation over the shelf, and coastal inundation in the nearshore-scale have typically been based on the non-dispersive nonlinear shallow water wave equations (NSWE; e.g., GeoClaw, George and LeVeque, 2008) or on dispersive Boussinesq-type such as FUNWAVE (e.g., Shi et al., 2012; Kirby et al., 2013) or non-hydrostatic wave equations such as NHWAVE (e.g., Ma et al., 2012; Tappin et al., 2014; Grilli et al., 2019). Modeling studies of tsunami propagation in the ocean with and without dispersion have indicated that, even for co-seismic tsunamis, frequency dispersion effects can accumulate to a sufficient degree to change waveforms, altering the spatial distribution of wave elevations and coastal inundation (Ioualalen et al., 2007; Horrillo et al, 2012; Kirby et al., 2013; Glimsdal et al. 2013; Zhou et al. 2012; Kirby, 2016). Due to wave dispersion and nonlinearity, tsunami wave crests often evolve into undular bores (a.k.a., dispersive shock waves) as they approach the shoreline, an effect which may significantly increase tsunami impact (i.e., currents and forces) on coastal structures (Madsen et al., 2008; Schambach et al., 2019). For landslide-generated tsunamis, wavelengths are relatively shorter, and thus wave dispersion effects cannot be neglected (e.g., Ma et al., 2012; Grilli et al. 2015, 2017, Schambach et al., 2019). As shown in the above studies, the magnitude of dispersive effects at given locations is a priori unknown; hence, it can only be estimated by performing simulations with a dispersive models for each specific event, whether hypothetical, historical or in real time. With this realization, in the last decade, modelers have gradually acknowledged the need for using a dispersive wave model to accurately assess tsunami hazard, especially nearshore effects.

Although some models use irregular grids or adaptive mesh refinement, the traditional way for carrying out multi-scale tsunami modeling has been to use nested grids, either with a one-way nesting or two-way nesting method. The grid nesting method is usually performed by nesting a fine grid within a coarse grid in a two- or multi-grid system with the hierarchical structure from coarser (lower-level) to finer grids (upper-level). In a one-way nesting, the model at an upper-level is forced by the boundary conditions obtained from the output of the lower-level model. There is no feedback from the upper-level grid to the lower-level grid. The nesting process can be done offline manually by running the model from the lower-level grid to the upper-level grid without an additional interface developed in the model. Kirby et al. (2013), Tappin et al. (2014), Schambach et al. (2019, 2020), and Nemati et al. (2019), for instance, are recent examples of using many levels of one-way nested spherical and/or





Cartesian grids, with FUNWAVE and/or NHWAVE, varying from a few meters or tens of meters nearshore, to 1 or 2 arc-min in the deep ocean. In a two-way nesting, the procedure to force the upper-level grid model is the same as the one-way nesting, but the feedback from the coarse grid to the fine grid is taken into account by updating the coarse grid solution with the fine grid solution. To achieve this, the calculations at all grid levels have to perform simultaneously. To this effect, an interface to handle the interactions between the nested grids has to be developed, which involves a significant programming effort.

Multi-scale tsunami modeling may also be carried out using adaptive mesh refinement (AMR). In an AMR model, the calculations at all grid levels have to perform simultaneously, in which the grid resolution is adaptively refined as a function of chosen features of the flow field, such as a high spatial gradient in the solution. AMR can be implemented using either an unstructured (e.g., Sleigh et al., 1998, Skoula et al., 2006) or block-structured scheme (Berger and Oliger, 1984; Berger and LeVeque, 1998; Liang, 2012). The latter is very similar to the two-way grid nesting mentioned above, except that the grid refinement is processed dynamically rather than prescribed using static sub-domains in the traditional two-way nesting framework. Over the last decade, the AMR technique has found increasing use in publicly available codes (see the review paper by Dubey et al., 2014). In tsunami applications, the AMR technique has been used in the NSWE-based models such as Geo-Claw (George and LeVeque, 2008, Watanabe et al., 2012, Arcos and LeVeque, 2014). For Boussinesq-type wave models, however, the higher-order numerical schemes and tridiagonal matrices, which are derived on a structured grid system, make it challenging to implement a quadtree-structured AMR; although a block-structured AMR is relatively easier to implement, its efficiency may be penalized by the large data management and computational costs when solving the complex nonlinear dispersive equations at multi-grid levels. Therefore, the AMR technique has rarely been applied to solving Boussinesq-type wave equations.

In practical applications of multi-scale tsunami modeling using a dispersive wave model, the traditional multi-grid one-way nesting approach has proved efficient and accurate when focusing on nearshore effects, provided the nearshore grid refinement ratio (i.e., ratio of discretization size from one nested grid to the next) is 4 or better. As the coastal area of interest is usually predetermined when setting up the model, the grid refinement can be generated at the beginning and remain unchanged throughout the entire simulation. Besides other applications mentioned above, a typical recent example is Tehranirad et al.'s (2020) FUNWAVE simulations of the far-field effects of the Tohoku-Oki 2011 tsunami in



Crescent City Harbor, California, using a nested grid system including the ocean basin,

regional, and nearshore harbor domains, as shown in Fig. 1. The basin-scale grid has a

2 arc-min resolution, covering the entire Pacific Ocean; the nested grids are then specified

in five levels along the U.S. west coast, with a hierarchical structure from a resolution

of 16 arc-sec to 1/6 arc-sec, downsizing towards the Crescent City Harbor domain. The

fully nonlinear Boussinesq model, FUNWAVE-TVD (Shi et al., 2012; Kirby et al., 2013)

is used in each individual grid with a one-way nesting scheme performed by applying the

boundary conditions obtained from a lower-level grid model. While this nesting process is

straightforward, it involves considerable post-processing effort to manipulate and interpolate

results from one level of nested grid to prepare data for simulating the next level grid. In

addition, the one-way nesting scheme may cause inconsistencies between different grids due

to wave reflection at each model boundaries.

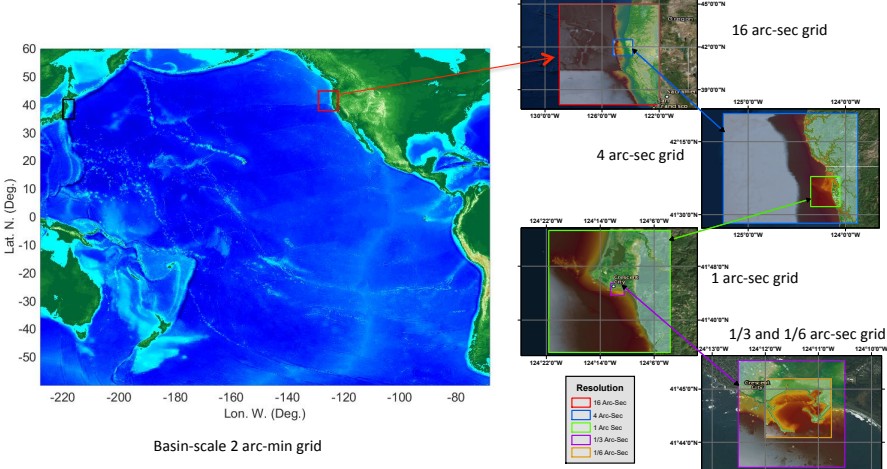

**Fig. 1.** The nested grids in the simulation of Tohoku-Oki 2011 tsunami impact on Crescent City Harbor, Oregon (Tehranirad et al., 2020). The nesting process scales down the grid resolution from 2 arc-min in the ocean basin domain to 1/6 arc-sec in the harbor domain. The same simulation was conducted using the present model nesting framework in section 4.4.



<sub>121</sub>    Yamazaki et al. (2010) implemented two-way nesting method in their dispersive depth-integrated, non-hydrostatic wave model for tsunami applications. The nesting model framework was based on a block-structured scheme with multiple prescribed nested grids. They used this model to simulate the 2009 Samoa tsunami and the coastal inundation caused in Pago Pago harbor, and reported good efficiency and accuracy of the two-way nesting model framework. However, it is not clear whether this two-way nesting scheme was parallelized and how the mega-data structure was handled in the nesting framework. Recently, Chakrabarti et al. (2017) implemented the fully nonlinear and dispersive Boussinesq model FUNWAVE-TVD in the block structured AMR framework CACTUS, which has been widely used in the field of astrophysics (Löffler et al., 2014). They showed that shallow water waves could be simulated at higher resolution, with a reasonable computational cost, which also allowed using an improved higher-order representation of the vegetation drag force. However, in this application, the nested grids were statically prescribed, to reduce the computational cost from using dynamically adapted grids with a Boussinesq-type model. In addition, the CACTUS-based version of FUNWAVE-TVD relies on a specific library package and configuration (Oler et al., 2016), limiting its general applications in the large user community.

<sub>137</sub>    There are significant challenges implementing an AMR and two-way multi-grid nesting framework in a parallel computing environment. Load balance, communication between parent and child grids, and mega-data management are major issues in Message Passing Interface (MPI)-based programs. Load balance is important for CPU scaling, in terms of synchronization of solutions across refinement levels. Dubey et al. (2014) reviewed load balancing methods in several public-domain AMR packages and pointed out the difficulties in achieving workload balance in the AMR framework. In a parallel multi-level grid system, the parent-child grid communication is also critical to modeling efficiency. Strategies to build direct communication between multi-level grids cross-ranks can be found in many AMR packages (Dubey et al., 2014). A multi-level grid system makes the meta-data management more complex, especially for tree-structured data. Finally, it is important to optimize the amount of meta-data replication according to both the communication cost and memory cost (Dubey et al., 2014).

<sub>150</sub>    The scope of the present work is to develop a multi-grid nesting framework for the Boussinesq-type wave model FUNWAVE-TVD, a widely-used public domain model in the nearshore and tsunami research community. FUNWAVE was initially developed by Kirby et al. (1998) based on the fully nonlinear Boussinesq equations derived by Wei et al. (1995).



The development of the Total Variation Diminishing (TVD) version of the model was motivated by a growing demand for phase-resolving modeling of nearshore waves and coastal inundation during storm or tsunami events. The model comprises a Cartesian mode (Shi et al., 2012) and a spherical mode (Kirby et al., 2013), an appropriate mode can be selected according to applications. The code was parallelized using the domain decomposition method based on MPI, for CPU-based High-Performance-Computing (HPC) clusters, and the GPU-accelerated program for single- and multi-GPU systems (Yuan et al., 2020).

The limitation of the prior CACTUS implementation of an AMR version of FUNWAVE-TVD to a specific HPC platform (Chakrabarti et al., 2017), motivates the development of a more platform-independent implementation of a two-way nesting scheme. The primary objective for the present development is to provide a generic interface, which can be used with any HPC platforms. The interface is developed separately from the core program and does not interfere with the main solver of FUNWAVE-TVD. Hence, the package of the combined interface and core program can be updated concurrently.

In the following, a brief description of the FUNWAVE-TVD model is given in section 2. Section 3 describes the two-way nesting interface, including the general algorithm, workload balance and flowchart of a MASTER program. Applications are presented in section 4. Section 5 provides a summary of the study.

## 2 FUNWAVE-TVD

FUNWAVE-TVD is a Boussinesq-type wave model discretized by a hybrid method combining finite-volume and finite-difference TVD-type schemes. The model was developed in both the Cartesian coordinates (Cartesian mode, Shi et al., 2012) and spherical coordinates (Spherical mode, Kirby et al., 2013). The Cartesian mode solves the fully nonlinear Boussinesq equations, initially derived by Wei et al. (1995), with the second-order correction of vertical vorticity by Chen (2006) and the moving reference level of Kennedy et al. (2001). The Spherical mode solves the weekly nonlinear, weakly dispersive Boussinesq equations in spherical coordinates (Kirby, et al., 2013). In tsunami applications, where nearshore waves are expected to be strongly nonlinear, a combination of deep water spherical and nearshore Cartesian grids has often been used in the one-way coupling nested grid framework (e.g., Grilli et al., 2013, 2015, 2017; Schambach et al., 2019, 2020; Tappin et al., 2014). Here, we





provide a brief summary of the governing equations, numerical schemes, and parallelization method.

### 2.1 Conservative form of Boussinesq equations in the Cartesian and spherical coordinate systems

Although the sets of equations in Cartesian and spherical coordinate systems are different, the two FUNWAVE modes were developed within the same numerical framework and using the same TVD-type solver. The combined form of the Boussinesq equations in the two coordinate systems can be written as:

$$\frac{\partial \mathbf{\Psi}}{\partial t} + \nabla \cdot \mathbf{\Theta}(\mathbf{\Psi}) = \mathbf{S}, \tag{1}$$

where $\mathbf{\Psi}$ and $\mathbf{\Theta}(\mathbf{\Psi})$ are the vector of conserved variables and the flux vector function, respectively, given by:

$$\mathbf{\Psi} = \begin{pmatrix} H \\ U \\ V \end{pmatrix}, \qquad \mathbf{\Theta} = \begin{pmatrix} S_p P\mathbf{i} + Q\mathbf{j} \\ \left[\frac{S_p P^2}{H} + \frac{1}{2}S_p g(\eta^2 + 2\eta h)\right]\mathbf{i} + \frac{PQ}{H}\mathbf{j} \\ \frac{S_p PQ}{H}\mathbf{i} + \left[\frac{Q^2}{H} + \frac{1}{2}g(\eta^2 + 2\eta h)\right]\mathbf{j} \end{pmatrix}, \tag{2}$$

where $(P,Q)$ are the horizontal volume fluxes:

$$(P,Q) = H(\mathbf{u}_\alpha + \bar{\mathbf{u}}_2), \tag{3}$$

where $H = h + \eta$ with $h$ the water depth and $\eta$ the surface elevation, $\mathbf{u}_\alpha$ is the horizontal velocity vector at a reference depth $z_\alpha$, $\bar{\mathbf{u}}_2$ is the depth-averaged second-order horizontal velocity of $O(\mu^2)$, in which $\mu$ is the dimensionless parameter quantifying the magnitude of wave dispersion. The velocity components $(U,V)$ combine $\mathbf{u}_\alpha$ and the time derivative dispersive terms $\mathbf{V}_1$:

$$(U,V) = H(\mathbf{u}_\alpha + \mathbf{V}_1). \tag{4}$$

The velocity $\mathbf{u}_\alpha$ is obtained by solving a system of equations with a tridiagonal matrix formed by (4).

In (2), $S_p$ is the spherical coordinate correction factor defined for the spherical mode as:

$$S_p = \frac{\cos \theta_0}{\cos \theta}, \tag{5}$$

in which $\theta$ and $\theta_0$ are the latitude and the reference latitude, respectively (see, Kirby et al., 2013). For the Cartesian mode, $S_p = 1$. The last term $\mathbf{S}$ in (1) contains the Boussinesq




source terms, which are detailed in Shi et al. (2012) for the Cartesian mode and Kirby et al. (2013) for the spherical mode.

## 2.2  Numerical schemes

In FUNWAVE-TVD, the HLL Riemann solver with the fourth-order accurate MUSCL-TVD scheme (Erduran et al., 2005) was implemented to discretize the leading order spatial derivative terms of the equations, while the dispersive terms were discretized by a second-order centered finite difference scheme. Choi et al. (2018) compared the performance of the MUSCL-TVD, WENO and MLP schemes in FUNWAVE-TVD, and showed that the MUSCL-TVD scheme with a van-Leer limiter provides an accurate and stable solution in long-term simulations.

For time stepping the equations, FUNWAVE uses the third-order Strong Stability-Preserving (SSP) Runge-Kutta scheme (Gottlieb et al., 2001), with an adaptive time stepping based on the Courant-Friedrichs-Lewy (CFL) condition prescribed as:

$$\Delta t = C_{\text{cfl}} \min \left( \min \frac{\Delta x}{|u_\alpha + \sqrt{gH}|}, \min \frac{\Delta y}{|v_\alpha + \sqrt{gH}|} \right), \tag{6}$$

where $C_{\text{cfl}}$ is the Courant number, and $\Delta x$ and $\Delta y$ are grid sizes in the $x$ and $y$ directions, respectively.

Although the conservative equations (1) are solved explicitly using the HLL Riemann solver, a system of tridiagonal matrix equations derived from (3) needs to be solved to get the velocity at the reference level, which is done with Thomas' algorithm (Naik et al., 1993).

Various boundary conditions were implemented in the model, including a wall boundary condition, wave periodic boundary condition, wavemaker boundary condition, and absorbing or partially absorbing boundary conditions. The wall boundary condition is the main boundary condition, dealing with wither full wave reflection or a moving shoreline. Ghost cells are used in the grid to implement a mirror boundary condition.

## 2.3  Parallelization

The CPU code uses a domain decomposition technique to subdivide the problem into multiple regions and assign each subdomain to a separate processor core. Each subdomain region contains an overlapping area of ghost cells, which is three rows deep, as required by



the fourth-order MUSCL-TVD scheme. MPI with non-blocking communication is used to

exchange data in the overlapping region between neighboring processors. The tridiagonal

matrices are solved using the parallel pipelining tridiagonal solver described in Naik et al.

(1993).

Data exchanges between neighboring subdomains are conducted through the ghost cells

at every Runge-Kutta time step. To increase the model efficiency, the values of dispersive

terms, in addition to the major variables $(\eta, P, Q)$, are also exchanged at the ghost cell

boundaries.

## 3 Two-way nesting interface

As mentioned in Section 1, the goal of the development here is to build an interface,

which can be used as a MASTER program to couple the sub-models with different grid

resolutions in a nested, two-ways, interactive manner. This way, FUNWAVE-TVD can be

used either stand-alone on a single grid or in a multi-grid nested system.

### 3.1 Algorithm

For simplicity, we consider first a two-nested grid system containing the parent grid
$\Omega_0$ and the child grid $\Omega_1$ as shown in Fig. 2. The parent grid has a larger grid size, $\Delta x_0$,
while the child grid has a smaller grid size, $\Delta x_1$. The grid refinement ratio is thus defined
as, $s = \Delta x_0 / \Delta x_1$. The boundary of the child grid is denoted by $\Gamma$, which has ghost cells.
Following the general procedure for two-way nesting, such as detailed in Debreu and Blayo
(2008), the partial differential equation (1) can be rewritten as:

$$\frac{\partial \mathbf{\Psi}}{\partial t} = L(\mathbf{\Psi}), \tag{7}$$

where $L(\mathbf{\Psi}) = \mathbf{S} - \nabla \cdot \mathbf{\Theta}(\mathbf{\Psi})$, represents a general operator. The equation is discretized in
$\Omega_0$ and $\Omega_1$ grids, by:

$$\frac{\partial \mathbf{\Psi_0}}{\partial t} = L_0(\mathbf{\Psi_0}), \qquad \frac{\partial \mathbf{\Psi_1}}{\partial t} = L_1(\mathbf{\Psi_1}), \tag{8}$$

respectively, where $L_0$ and $L_1$ denote the discretized form of the same operator $L$ at a

different resolution. In the two-way nesting framework, the child grid solution is driven by

the lateral boundary conditions along $\Gamma$, while the parent grid is updated using the child

grid solution. Noting that both of the procedures need interpolation/mapping processes,

we define the interpolators, $I_s$ and $I_t$, and the restriction operator, $R$. $I_s$ and $I_t$ perform

interpolations in space and time, respectively, at $\Gamma$, and $R$ performs the mapping from the





child grid solution to the parent grid. Assuming the grid refinement factor, $s$, equals the

time refinement factor based on the CFL condition (6), the two-way nesting can be described

by the following pseudo code:

$$\boldsymbol{\Psi_0}^{n+1} = L_0(\boldsymbol{\Psi_0}^n)$$

$$\text{loop} \quad i = 1 \text{ to } s$$

$$\boldsymbol{\Psi_1}^{n+\frac{i}{s}} = L_1(\boldsymbol{\Psi_1}^{n+\frac{i-1}{s}})$$

$$\text{with} \quad \boldsymbol{\Psi_1}^{n+\frac{i}{s}}|_\Gamma = I_t[I_s(\boldsymbol{\Psi_0}^n), I_s(\boldsymbol{\Psi_0}^{n+1})]$$

$$\text{end loop}$$

$$\boldsymbol{\Psi_0}^{n+1} \in \Omega_1 = R(\boldsymbol{\Psi_1}^{n+1}).$$

Three ghost cells are used along $\Gamma$, which are required by the higher-order numerical schemes

used in the model.

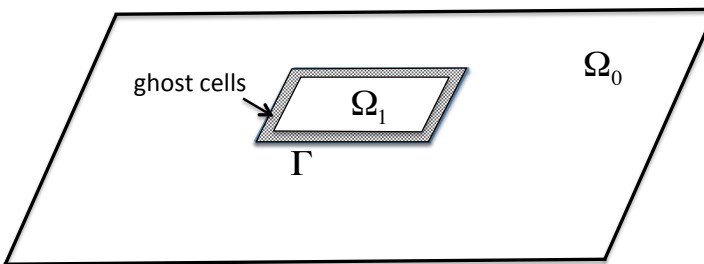

**Fig. 2.** Schematic drawing for two-way nesting method. The parent grid $\Omega_0$ has a coarse resolution grid while the child grid $\Omega_1$ is high resolution. Ghost cells (3 rows) are specified along the inter-grid boundary $\Gamma$.

### 3.2  Interpolator and restriction operator

In some two-way nesting methods used in 3D ocean models, the interpolators, $I_s$ and

$I_t$, and the restriction operator, $R$, are complex due to issues raised by mass/momentum im-

balance, barotropic/baroclinic mode splitting, and the staggered grid configuration (Debreu

et al., 2012). To ensure mass and momentum conservation during the two-way nesting, a

correction may be needed at the nesting boundaries according to specific numerical schemes.



²⁵²    This especially occurs in a nesting scheme using discretizations of the unconservative forms

²⁵³    of mass and momentum equations, based on finite differences, because the flux of mass or

²⁵⁴    momentum is expressed by a nonlinear term. Hence, typically, the mass flux, $(h + \eta)u$, is

²⁵⁵    no longer conserved when performing a linear interpolation individually for $\eta$ and $u$ at a

²⁵⁶    nesting boundary.

²⁵⁷    FUNWAVE-TVD is based on the conservative forms of mass and momentum equations,

²⁵⁸    in which advection is performed using the finite volume method (Shi et al., 2012). The latter

²⁵⁹    makes it possible to use a linear or doubly-linear interpolator in the nesting method, without

²⁶⁰    changing the conservative property of the equations. In the AMR application of a NSWE

²⁶¹    model, Liang (2012) demonstrated the conservative property of the linear operator used in

²⁶²    the finite volume Godunov-type scheme and, later, pointed out that the operator preserves

²⁶³    both mass conservation and the C-property (i.e. conservation property) as the wetting-

²⁶⁴    drying process envolves in the grid nesting. FUNWAVE-TVD uses a finite volume scheme

²⁶⁵    similar to Liang's (2012) and Liang et al.'s (2105) and, therefore, its conservative property

²⁶⁶    should be maintained when applying a linear operator. Unlike Liang (2012), who used a

²⁶⁷    second-order scheme, FUNWAVE-TVD applies a higher-order Godunov-type scheme, hence

²⁶⁸    ghost cells must be used along nesting boundaries.

²⁶⁹    Consequently, a doubly-linear interpolation is applied to ghost points in the child do-

²⁷⁰    main, using values from the parent grid. Thus, at a ghost point $(X, Y)$ in the child do-

²⁷¹    main, which is surrounded by four points, $(x_{ij}, y_{ij})$, $(x_{i+1,j}, y_{i+1,j})$, $(x_{i+1,j+1}, y_{i+1,j+1})$,

²⁷²    $(x_{i,j+1}, y_{i,j+1})$, in the parent domain, a given variable $\varphi_1$ is interpolated as:

$$\varphi_1 = \left[t\,\varphi_{ij} + (1-t)\varphi_{i+1,j}\right]s + \left[t\,\varphi_{i,j+1} + (1-t)\varphi_{i+1,j+1}\right](1-s), \tag{9}$$

²⁷³    where,

$$t = \frac{x_{i+1,j} - X}{x_{i+1,j} - x_{ij}}, \quad s = \frac{y_{i,j+1} - Y}{y_{i,j+1} - y_{ij}}, \tag{10}$$

²⁷⁴    where $\varphi_{ij}$, $\varphi_{i+1,j}$, $\varphi_{i,j+1}$, $\varphi_{i+1,j+1}$ are values of the variable in the parent domain.

²⁷⁵    The restriction operator uses linear averaging, which guarantees the conservation of

²⁷⁶    mass and momentum.





### 3.3 Workload balance and data management

The MPI parallelization of FUNWAVE-TVD uses a 2D Cartesian topology for the domain-decomposition, which subdivides the computational domain into a 2D grid, each cell of which is assigned to a processor. The size of the global arrays is not necessarily divisible by the number of processors, but an evenly divisible configuration results in a perfectly equal workload. To ensure workload balance in computations involving multi-grid levels, we used the same domain-decomposition algorithm on all grid levels with the same number of processors. This algorithm is especially efficient for block-structured or patch-structured nesting schemes, as described in Debreu et al. (2012).

Fig. 3 gives an example of the domain-decomposition and communication at two-grid levels in a system of 9 processors. Both the parent computational domain and the child domain are decomposed evenly into a 3 × 3 grid, according to the standard 2D Cartesian virtual topology used in the MPI library, with ranks named ID=1, 2, ..., 9. The communications between the parent and child grids are straightforward without a data-gathering process using an additional processor or a global array. In this example, along the west boundary of the child grid, the processors with ID=1, 2 and 3 communicate directly with processors with ID = 4 and 5 in the parent grid. The parent-child proximity is pre-calculated at the beginning of the model run, and hence will not require additional computational cost. The same strategy is used by the restriction process.

As mentioned in the introduction, our goal in development is to make a generic grid nesting interface without altering the main FUNWAVE-TVD code. To achieve this, we treated the main program of the original model as a kernel, which performs computations at all grid levels. The kernel is called from the MASTER program, which manages the time sequencing and nesting processes. A strategy of shared array allocation is used, whereby the arrays are allocated with the maximum dimension of all grids at the initialization stage and grids at all levels share the same memory allocations. New arrays are created only for the storage of boundary conditions at all levels. There is no additional data structure implemented in the meta-data management.



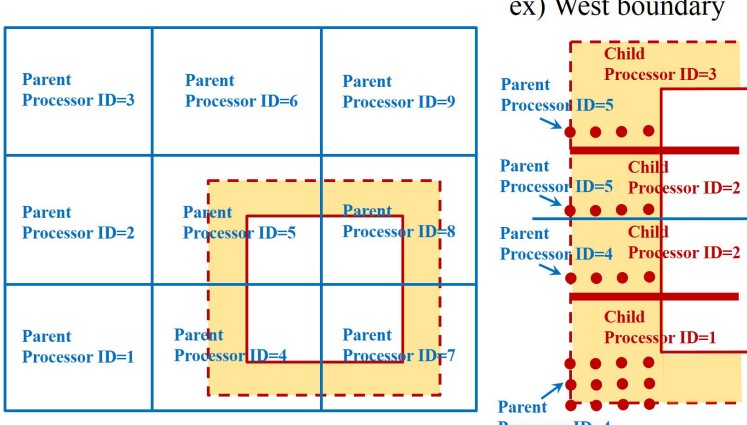

**Fig. 3.** Decomposed domain by MPI and the relation between parent processor ID and child processor ID.

### 3.4 Flowchart

Fig. 4 summarizes the flowchart of the MASTER program. After the MPI initialization, the program reads input data, including model parameters needed in the original model and nested grid information. As mentioned earlier, array allocation and initialization are performed based on the maximum dimension of all grid levels. Additional arrays for the storage of boundary conditions are also allocated at this stage. Then the program starts the main time loop based on time stepping of the background (first level parent) grid. The calculations at each grid level are conducted hierarchically inside the main time loop, with a time steps based on the grid refinement ratio $s$. At each grid level, the model is assigned by the initial condition (solution at last time level) and boundary conditions obtained from the $I_s$ and $I_t$ interpolation processes. Then the core FUNWAVE-TVD program is run at the grid level and stores boundary values for the child grid. All parent grids are updated based on the child grid results through the $R$ process, after all subgrid levels computations are completed.



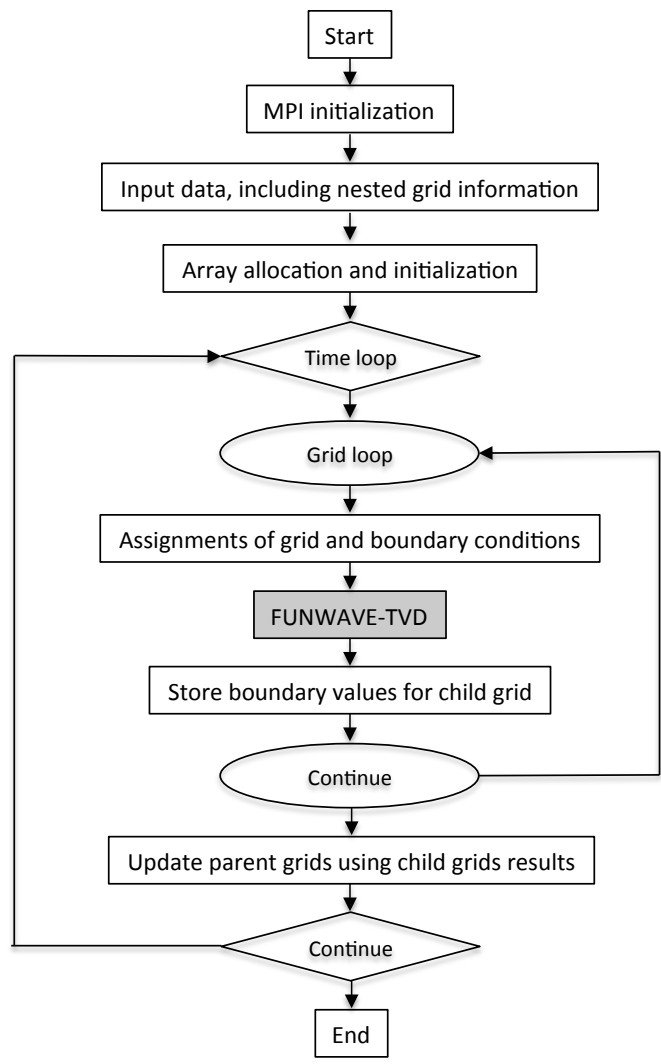

**Fig. 4.** Flowchart of the two-way nesting interface.

## 4 Applications

Hereafter we test our new two-way coupled nested grid solution with FUNWAVE-TVD on a series of standard idealized or benchmarking applications, and then on the Tohoku 2011 tsunami case study discussed earlier.





### 4.1 Evolution of an initial rectangular-shaped hump

The evolution of waves generated from an initial arbitrary rectangular-shaped hump on the free surface is used to test the consistency of the multi-grid nesting system and effects of the higher-resolution resulting from the grid refinement. As shown in Fig. 5, a 100 m × 100 m hump with an elevation of 1 m is specified, with no initial velocity, at the center of a 500 m × 500 m rectangular domain with a 5 m water depth. Wall boundary conditions (fully reflective) are specified at the four boundaries of the domain. The initial still water level can thus be defined as:

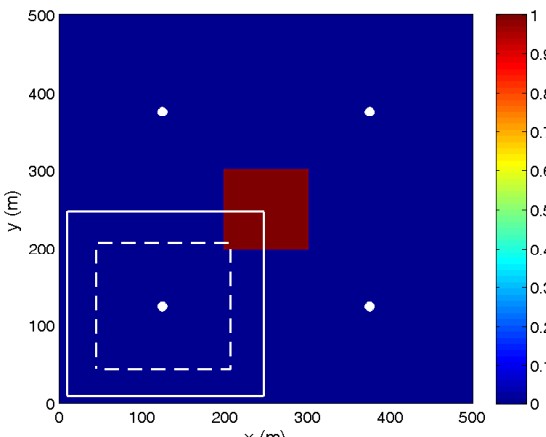

**Fig. 5.** Wave generation from an initial 1 m elevation still water hump. The parent Grid 1 covers the entire domain and solid/dashed lines mark the boundary of Grid 2/3. Bullets mark locations of numerical wave gauges for comparing free surface elevations. Color represents the initial surface elevation in meter.

$$\eta(x,0) = \begin{cases} 1.0, & 200 \leq x \leq 300, 200 \leq y \leq 300, \\ 0.0, & \text{elsewhere.} \end{cases}$$

The consistency and accuracy of the two-way nested grid algorithm is first assessed by defining a three-level nested grid system with identical grid resolution $\Delta x = \Delta y = 2.5$ m in Cartesian coordinates, hence a grid refinement ratio $s = 1$. Grid 1 is the background

—
—

—
—

—

—

—
—
—

—
—
—

—

—
—

—

—

—

—

—

—

—

—

—
—

—

—

—

—

—

—

—
—
—
—

—




parent grid and Grids 2 and 3 are nested grids located in $10.0 \leq x, y \leq 247.5$ and $30.0 \leq x, y \leq 207.5$, respectively. Because the refinement ratio is 1, the same numerical solution is expected whether nesting is used or not. To verify this, surface elevation time series were computed at 4 numerical wave gauges located at, $(x, y) = (125, 125)$, $(375, 125)$, $(125, 375)$, and $(375, 375)$ m (Fig. 5). The bottom left gauge is located within the two nested grids and its time series computed in Grid 3 is compared to those at the three gauges located in Grid 1. Because of the symmetry of the initial solution, results at the four gauges should be identical, which is verified in Fig. 6, hence assessing the consistency of the nested grid model.

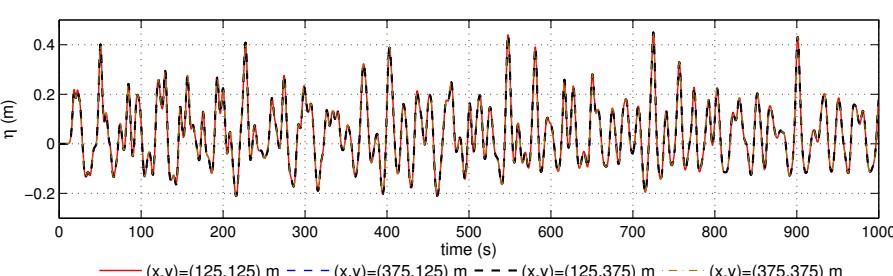

**Fig. 6.** Wave generation from an initial 1 m elevation still water hump. Comparison of surface elevation recorded at the four numerical wave gauges marked in Fig. 5. Nested grids have the same grid resolution of 2.5 m as the background parent grid.





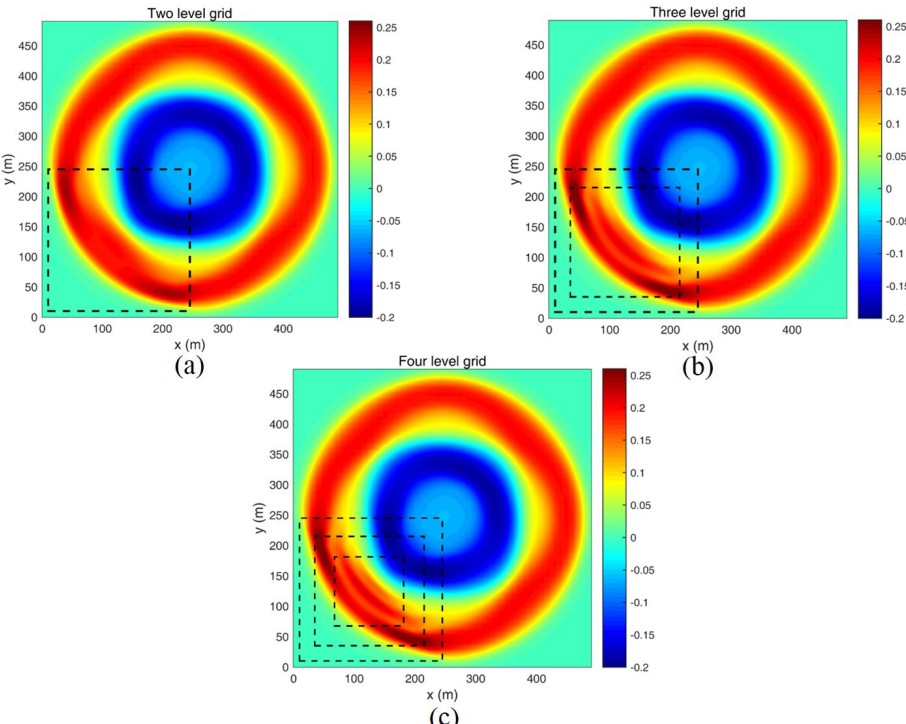

**Fig. 7.** Wave generation from an initial 1 m elevation still water hump. Snapshots of surface elevations computed at $t = 23$ s in: (a) two-level grids, (b) three-level grids, and (c) four-level grids, with discretization of 10, 5, 2.5, and 1.25 m, respectively ($s = 2$). Dashed lines denote the boundary of child grids.

Next, we examine the effects of grid refinement in a hierarchical nested grid system, for the same application. Here, the Grid 1 resolution is $\Delta x = \Delta y = 10$ m and Grid 2 and Grid 3 are nested within Grid 1, as before, but with grid resolutions of 5 m and 2.5 m, respectively. Grid 2 is located in $10.0 \leq x, y \leq 245$ and Grid 3 in $35.0 \leq x, y \leq 215$. An additional grid, Grid 4 was added within Grid 3, located in $67.5.0 \leq x, y \leq 181.25$ with a grid resolution of 1.25 m. Three computations were run using two- to four-level of nested grids, with surface elevations computed at $t = 23$ s shown in Fig. 7. Because wave dispersive effects are related to grid resolution, the solution in a finer grid is not exactly the same as in a coarser grid, resulting in asymmetric distributions of surface elevations in the figure. With a two-level nested grid system (Grids 1 and 2), Fig. 7a shows the appearance of sharper crests (dark





red) in Grid 2, as compared to the solution in Grid 1. As more levels of nested grids are

used, Figs. 7b and c show that shorter waves increasingly appear in the finer grids.

### 4.2 Wave refraction-diffraction over a shoal on a sloping bottom topography

Although the main targeted applications of our new two-way grid nesting model system

are tsunami simulations in multi-scale cases, the method can also be applied to the modeling

of ocean wave transformations in coastal areas. This is demonstrated here by simulating the

laboratory experiments of Berkhoff et al. (1982), for wave refraction-diffraction over a shoal

on a 1/50 sloping bottom topography, both rotated by 20° off the $y$-axis (Fig. 8). This

experimental dataset has served as a standard benchmark for assessing the accuracy and

performances of numerical wave models for simulating wave shoaling, refraction, diffraction,

and nonlinear dispersion. Shi et al. (2011) showed that the original version of FUNWAVE-

TVD accurately reproduces measured wave heights in this experiment.

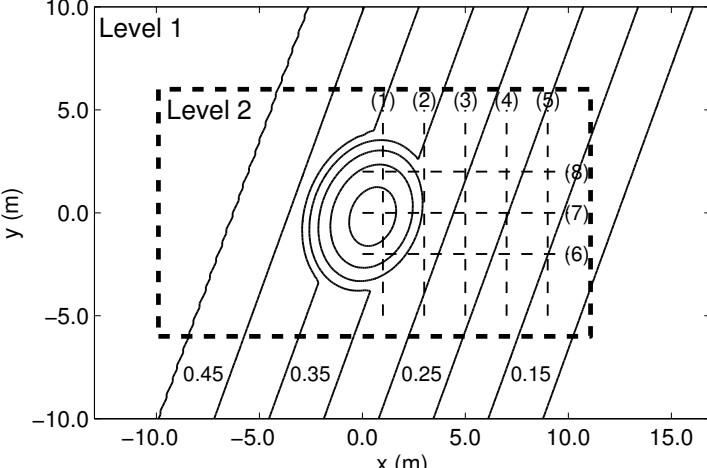

**Fig. 8.** Computational domain and bottom topography of Berkhoff et al.'s (1982) experiments of wave transformations over a tilted elliptical shoal on a sloping bottom. The bold dashed line box marks the area of nested Grid 2 (Level 2). Dashed lines (1) - (8) are transects for model/data comparisons.



Two numerical simulations were carried out for: (i) the original single grid model; and (ii) a two-level nested grid model (Fig. 8). Both models are set up in a rectangular domain with Cartesian coordinates, $-13\ m \leq x \leq 16.9\ m$ and $-10\ m \leq y \leq 10\ m$. The single grid model has grid resolutions of $\Delta x = 0.025$ m and $\Delta y = 0.05$ m. In the two-level grid model, Grid 1 is coarser with $\Delta x = 0.1$ m and $\Delta y = 0.2$ m resolutions, and the finer Grid 2 is nested in the region of $-9.9$ m $\leq x \leq 11.075$ m and $-6$ m $\leq y \leq 6$ m, with resolution $\Delta x = 0.025$ m and $\Delta y = 0.05$ m identical to those of the single grid mode, corresponding to a grid refinement ratio is thus $s = 4$. Total numbers of cells in Grid 1 and 2 are 30,300 and 202,440, respectively, which is much smaller than the 480,000 cells of the single grid model.

In both model setups, regular waves with a period $T = 1$ s and an amplitude $A = 4.64$ cm are generated, as in experiments, by a wavemaker located at $x = -10$ m. Sponge layers with a width of 2 m were specified on the left and right boundaries of both the single grid domain and Grid 1 in the nested grid model.



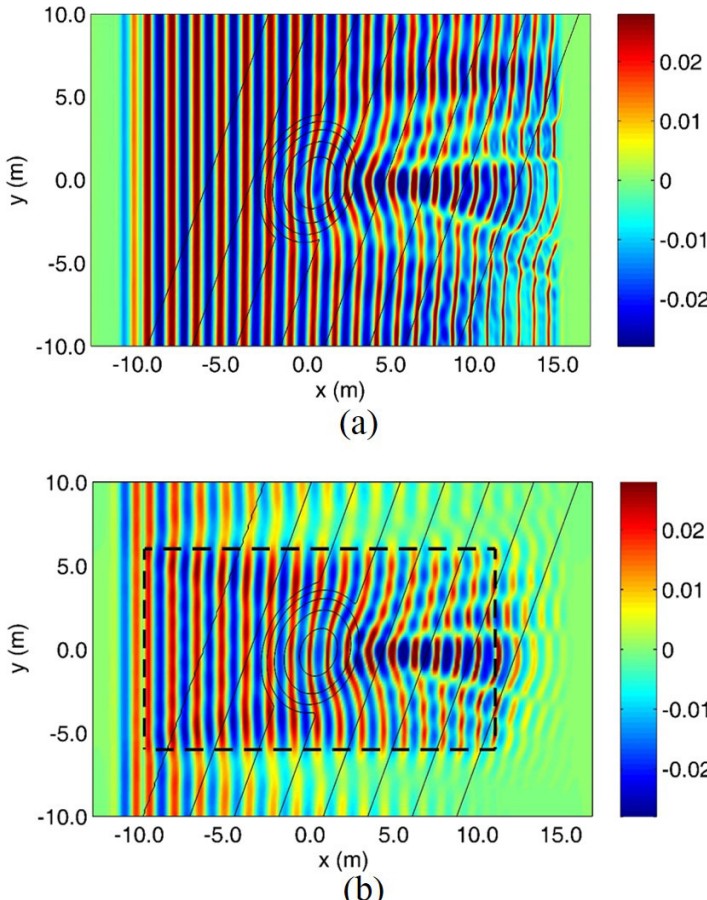

**Fig. 9.** Experimental benchmark of Fig. 8. Snapshots of free surface elevation (color scale in meter) at $t = 40$ s, computed in: (a) the single grid; and (b) the nested grid models.





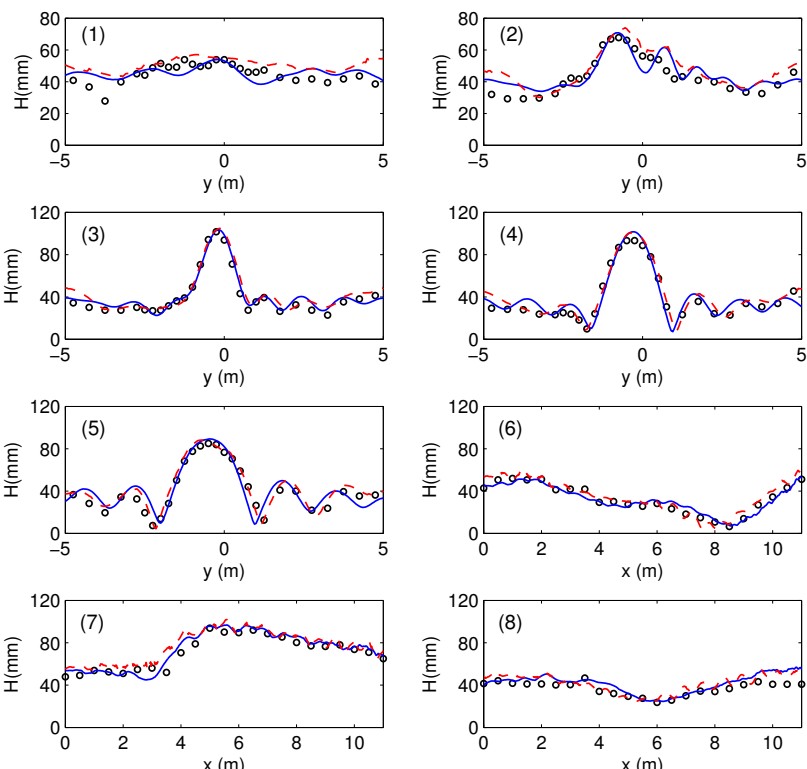

**Fig. 10.** Experimental benchmark of Fig. 8. Comparison of wave height distribution along transects (1)-(8) in: (o) experimental data and results of (dash) single grid, and (solid) nested grid model.

Fig. 9 shows snapshots of surface elevation computed at $t = 40$ s in both model setups. Compared to the single grid model, which uses the finest grid resolution over the entire domain, the nested grid model, which only uses it in Grid 2, shows that waves are numerically damped due to the coarse grid resolution used outside of Grid 2. Over the shoal and slope behind it, results in both the nested Grid 2 and the single grid show similar intense wave shoaling, refraction, and diffraction patterns. However, in the nested grid model, additional spurious wave diffraction effects can be seen around the lateral nesting boundaries due to the wave damping on the coarse grid side.

Fig. 10 shows comparisons of both model results with experimental data for the wave height variation along the transects marked in Fig. 8. For all transects, both the single



grid and nested grid model results agree well with the data. As expected from the spurious diffraction effects, compared to the single grid model, the nested grid model predicts slightly smaller wave heights at the ends of transects (1) – (5).

Regarding computational efficiency, in this application, the cost of the nested grid model is about 46.5 % that of the single grid model. It should be mentioned that this test is only for verification of the nested grid algorithm and is not a typical case for demonstrating the efficiency of the nested grid method.

### 4.3 Solitary wave runup on a shelf with an island

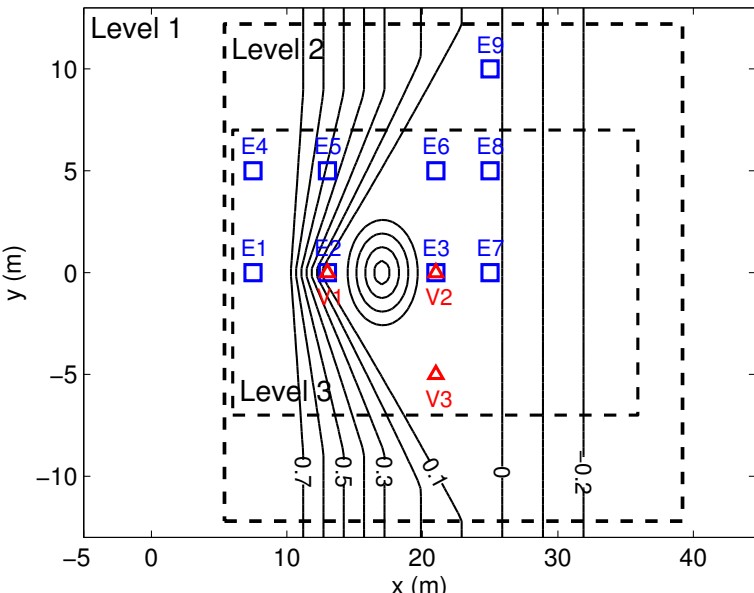

**Fig. 11.** Solitary wave runup on a shelf with an island (Lynett et al., 2010). Bathymetry contours (solid lines, in meters) and measurement locations in the computational domain. The parent Grid 1 (Level 1) covers the entire domain; blocks with dashed lines mark the boundaries of nested Grids 2 and 3 (Levels 2 and 3). Symbols mark locations of: (□) physical/numerical wave gauges E1–E9, and (△) velocimeters (ADVs), V1–V3.



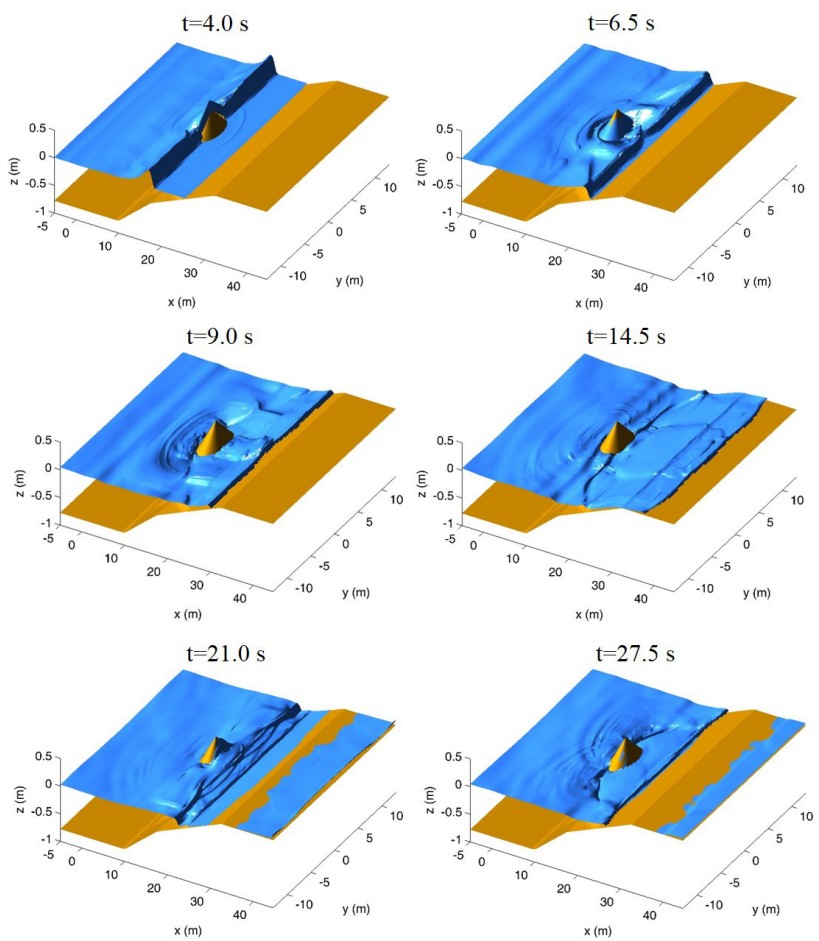

**Fig. 12.** Case of Fig. 11. Surface elevations simulated at $t = 4.0$, $6.5$, $9.0$, $14.5$, $21.0$ and $27.5$ s in nested grid system.




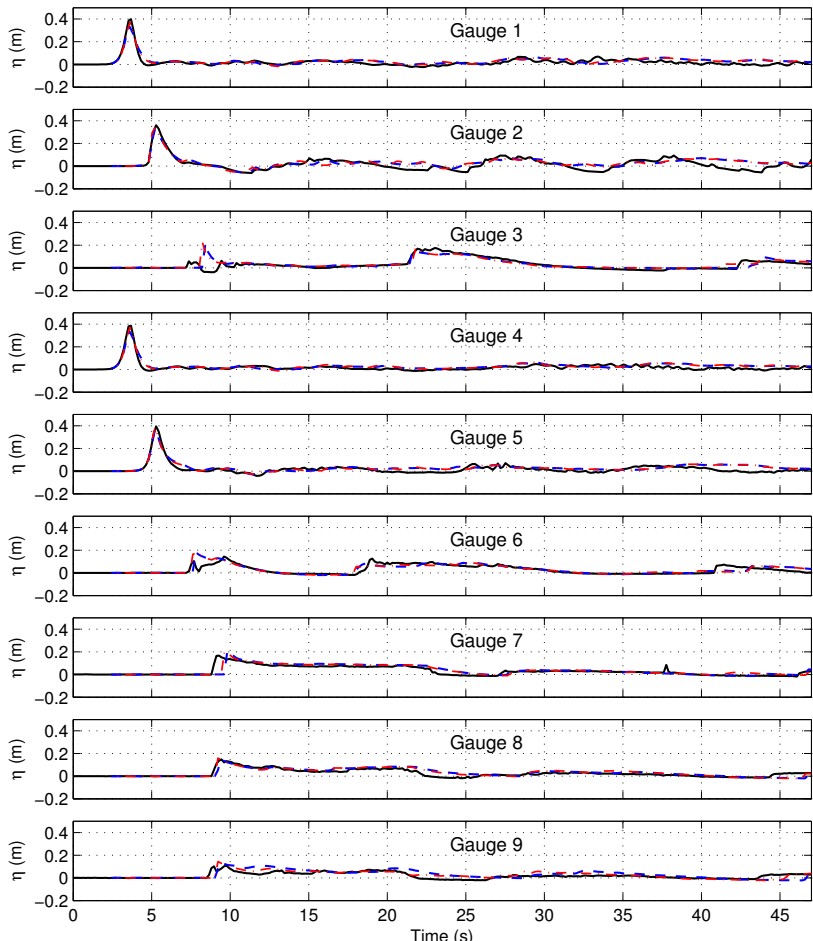

**Fig. 13.** Case of Fig. 11. Comparison of surface elevations at wave gauges in: (solid) experiments; (blue/red dash) present nested model/original FUNWAVE-TVD results.



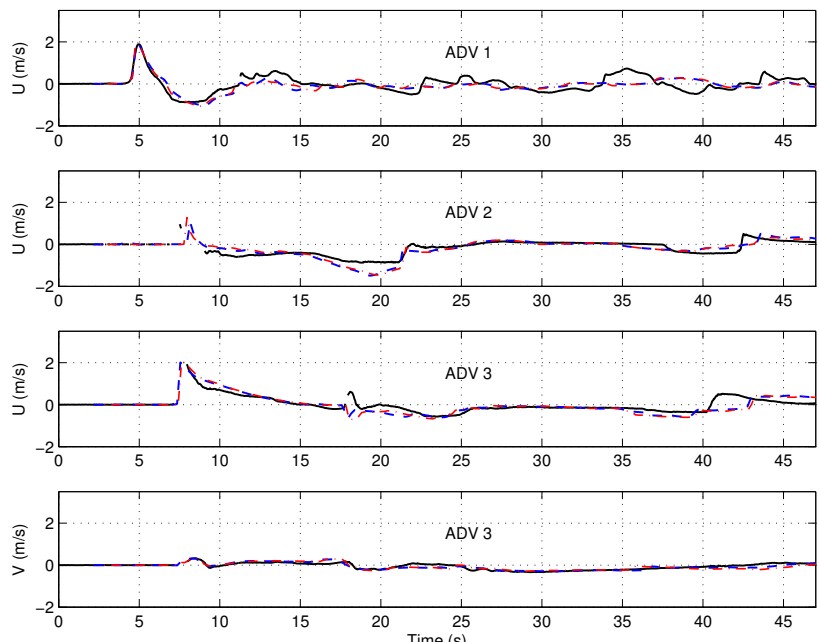

**Fig. 14.** Case of Fig. 11. Comparison of velocities at ADV locations in: (solid) experiments; (blue/red dash) present nested model/original FUNWAVE-TVD results.

A second experimental benchmark, for the runup of a solitary wave over a complex nearshore bathymetry with an island (Fig. 11), is simulated to assess the accuracy of the wetting and drying algorithm along the model shoreline, in a nested grid system. These experiments were performed in the large wave basin of Oregon State University's O.H. Hinsdale Wave Research Laboratory (Lynett et al., 2010). The 3D bathymetry was constructed in the 48.8 m long and 26.5 m wide basin, with a 2.1 m depth. It consists of a 1/30 plane slope connected to a triangular shelf with a conical island over the shelf (Figs. 11 and 12). Surface elevations were measured at nine locations using wave gauges (E1 – E9 in Fig. 11), and velocities were measured at three locations by Acoustic Doppler Velocimeters (ADVs) (V1 – V3 in Fig. 11). Details of the experiment can be found in Lynett et al. (2010). Shi et al. (2012) applied the original version of FUNWAVE-TVD to this case.

In the model simulations, a three-level nested grid system is set-up with Grid 1, Grid 2, and Grid 3 shown in Fig. 12. The model setup for Grid 1 is similar to Shi et al.'s (2012), except that grid resolution is coarser, with $\Delta x = \Delta y = 0.4$ m, versus 0.1 m in the original





Table 1: Grid information for the solitary wave experiment.

| Domain | $x$ range $(m)$ | $y$ range $(m)$ | mx $\times$ ny | $\Delta x$, $\Delta y$ $(m)$ |
|---|---|---|---|---|
| Level 1 | -5.0 $\sim$ 44.6 | -13.0 $\sim$ 13.0 | 66 $\times$ 125 | 0.4 |
| Level 2 | 5.4 $\sim$ 39.2 | -12.2 $\sim$ 12.2 | 123 $\times$ 170 | 0.2 |
| Level 3 | 6.0 $\sim$ 35.9 | -7.0 $\sim$ 7.0 | 141 $\times$ 300 | 0.1 |

model. The nested grids, Grid 2 and 3, have 0.2 m and 0.1 m resolution, respectively (hence, $s = 2$), and are centered in the middle of the domain where wetting and drying frequently occur due to the moving shoreline during runup. As measured in experiments, an incident solitary wave of height $H_o = 0.39$ m is specified in Grid 1, in the constant depth $h_o = 0.78$ m region on the left side of the model, from -5 m $< x <$ 5 m, with its crest initially located at $x = 0$. The initial solitary wave condition is based on Nwogu's extended Boussinesq equations (Wei, 1997). With $H_o/h_o = 0.5$ this represents a strongly nonlinear incident wave. A summary of the nested grid configuration is given in Table 1.

Fig. 12 shows snapshots of surface elevations simulated in the nested grid model, constructed using results from all grids, wherever the highest resolution results are available. Results show successively that wave breaking occurs at $t = 4.0$ s, edge wave collide behind the island at $t = 6.5$ s, a breaking bore forms at $t = 9.0$ s, with its front running-up and -down the upper slope and beach terrace, from $t = 9.0$ to 27.5 s. These are all quite complex processes that appear well-resolved in the nested grids.

Fig. 13 shows the comparison of model results with experimental data for surface elevations measured at the nine wave gauges (E1 – E9 in Fig. 11). Results from the single grid model (with grid resolution 0.1 m) are also plotted in the figure for comparison. Surface elevations simulated in the new nested grid model are quite close to those in the original single grid model, and both agree well with the experimental data. Slight differences between the nested grid and single grid models can be seen at Gauge 9, likely because this gauge is located in Grid 2, for which the resolution is lower than that in the single grid model; all the other gauges are located in Grid 3 which has the same resolution as the original single grid model.





Table 2: Grid parameters for the 2011 Tohoku-Oki tsunami simulation.

| Domain | Range of longitude (°E) | Range of latitude (°N) | mx × ny | Resolution (arc-min) |
|---|---|---|---|---|
| Level 1 | 132.0000 ∼ 292.0000 | -60.0000 ∼ 60.0000 | 2400 × 1800 | 4 |
| Level 2 | 205.0833 ∼ 238.4000 | 29.7500 ∼ 49.7333 | 2000 × 1200 | 1 |
| Level 3 | 221.7000 ∼ 236.6917 | 37.2000 ∼ 45.5250 | 1800 × 1000 | 1/2 |
| Level 4 | 232.9250 ∼ 236.2542 | 40.6750 ∼ 43.1708 | 800 × 600 | 1/4 |
| Level 5 | 234.8292 ∼ 235.9521 | 41.4750 ∼ 42.4313 | 540 × 460 | 1/8 |
| Level 6 | 235.6563 ∼ 235.9052 | 41.5833 ∼ 41.8531 | 240 × 260 | 1/16 |
| Level 7 | 235.7677 ∼ 235.8401 | 41.6844 ∼ 41.7359 | 140 × 100 | 1/32 |

Fig. 14 similarly compares time series of simulated and measured mean horizontal velocity at 3 ADVs (V1 – V3 in Fig. 11). Results from the nested grid model are all close to those of the original single grid model and both agree well with the data. We note that all ADVs are located in Grid 3, which has the same grid resolution as the original single grid model, hence results of both models are expected to be consistent.

### 4.4   Tohoku-Oki 2011 tsunami impact on Crescent City harbor, CA

As mentioned in the introduction, the multi-scale modeling of transoceanic tsunamis is a typical application of the nesting grid technique. Tehranirad et al. (2020) used the one-way nesting technique with six-level nested grids to simulate the impact of the Tohoku-Oki 2011 tsunami, particularly morphological changes, in Crescent City Harbor, CA. This harbor is known for its vulnerability to tsunamis due to wave-guiding effects caused by a ridge feature in the bottom topography of the Pacific Ocean (Grilli et al., 2013). During the 2011 tsunami, Crescent City Harbor experienced extensive damage caused by a significant inundation, but most of all strong currents induced within the harbor by successive long waves in the incoming tsunami wave train. Tsunami-induced oscillations of the harbor, and currents, were reported to have lasted for several days in the harbor (Wilson et al., 2012), due to nearshore edge waves associated with the tsunami event. When using that many levels of grids, the multi-scale modeling using the one-way nesting technique is particularly cumbersome, in terms of the manual post processing it involves. Hereafter, we repeat this



simulation using the new two-way nesting framework. Unlike the three earlier tests, which used the Cartesian mode, this test uses spherical coordinates.

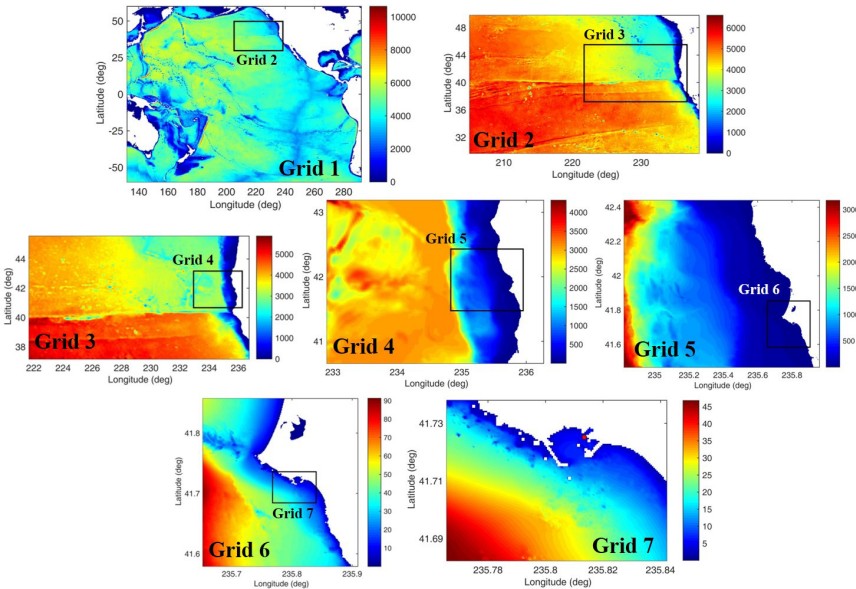

**Fig. 15.** Nested grid simulation of the 2011 Tohoku-Oki tsunami. Bottom topography (color scale in meter) and computational domains for Grids 1 to 7. Red circles in Grid 7 denote numerical wave gauge locations.

The nested grid system uses seven levels, with grid resolutions varying from 4 arc-min at the ocean basin scale to 1/32 arc-min around the harbor, and a nesting ratio $s = 2$. As shown in Fig. 15, with a high resolution of 1/32 arc-min in Grid 7 (or about 53 m), the model is able to resolve the harbor structures quite well. Following Tehranirad et al. (2020), the bathymetry used to define the model grids was constructed by combing 1 arc-min ETOPO-1 data (Amante and Eakins, 2009), 3 arc-second Coastal Relief Model (CRM) data (NGDC, 2003), and the local 10 m resolution tsunami DEM of Crescent City Harbor (Grothe et al., 2011). The tsunami was generated using the same source configuration as in Grilli et al. (2013) and Kirby et al. (2013). Model parameters were specified according to Tehranirad et al. (2020). Table 2 summarizes the locations, dimensions, and grid sizes of the nested grids.



Fig. 16 show snapshots of tsunami surface elevations computed in the basin-scale Grid 1 and the nested grids, Grid 5 and Grid 7, at $t = 11.2$ and 11.4 hr, when the water surface elevation within the harbor reaches its maximum and minimum levels, respectively (Fig. 17). The model shows the generation of edge waves propagating along the coast (Grid 5), which were not simulated in Tehranirad et al.'s (2020) one-way nesting computations. The two-way nesting is a more relevant technique to model waves propagating across nesting boundaries, without significant wave reflection from the boundaries.

Fig. 17 compares the modeled surface elevation with the data measured at the gauge location within the harbor (red circle in Grid 7 in Fig. 15). Following Tehranirad et al. (2020), the model result were shifted by 8 minutes backward to compensate for the time delay identified in earlier studies, which was possibly caused by compressibility and earth elasticity effects (Allgeyer and Cummins, 2014, Wang, 2015, Abdolali and Kirby, 2017, Abdolali et al., 2019). Overall, the model shows a good agreement with the data, although the largest wave crests are slightly over predicted.




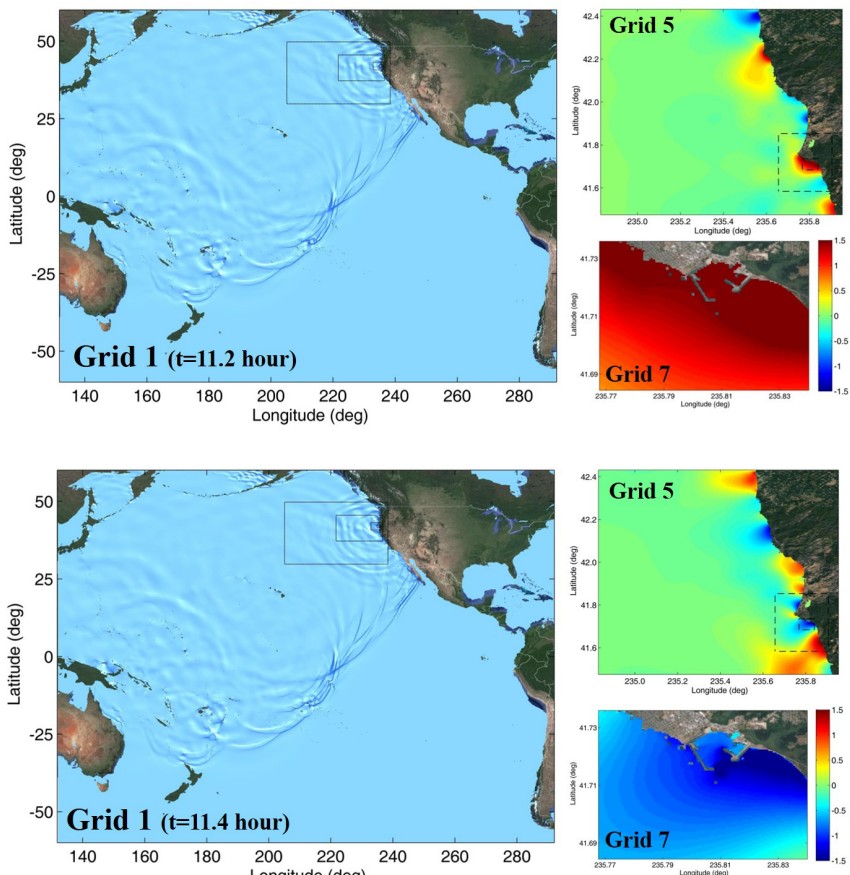

**Fig. 16.** Same case as Fig. 15. Snapshots of tsunami surface elevations simulated in Grid 1, Grid 5, and Grid 7 at $t = 11.2$ and $11.4$ hr.

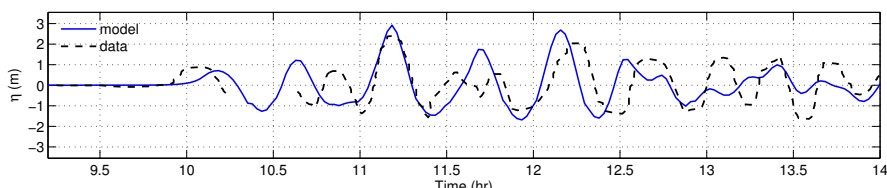

**Fig. 17.** Same case as Fig. 15. Comparison between model result and measured data inside the Crescent City harbor.





## 5 Conclusions

The main goal of this study was to develop a multigrid nesting interface for the Boussinesq wave model, FUNWAVE-TVD, which can be used as a MASTER program to manage time sequencing and nesting processes, and make it both easier and more accurate and efficient performing multi-scale tsunami simulations. The background model couples a series of submodels with grid refinement in a hierarchical manner. Unlike other AMR-type models, the new modeling framework does not alter the original solver, and hence FUNWAVE-TVD can still be used as a stand-alone program for each individual grid.

The nesting algorithm performs a two-way coupling between the parent and child grids. The child grid is driven by the boundary conditions provided by the parent grid. Linear interpolators are performed both in time and space at the ghost cells of nesting boundaries. The parent grid is updated with results from the child grids using a linear restriction operator. No correction of mass and momentum is needed during the nesting process because of the use of conservative forms of mass and momentum equations.

Workload balance is handled by an equal-load scheme, which performs the same domain-decomposition algorithm on all grid-levels using the same number of processors, guaranteeing equal CPU-load over the entire computation. Communication between the parent and child grids is direct without a data-gathering process. The parent-child proximity is pre-calculated at the beginning of the model run and, hence, does not cause additional computational cost. A strategy of shared array allocations is used in data management. Grids at all levels share the same memory allocations, and no additional memory allocation is required, allowing for a large number of nesting levels to share the same memory allocation.

The nested grid model was verified on four applications, three of which are standard benchmarks and one is a tsunami case study. The numerical test of wave evolution from a rectangular hump examined the consistency and general performance of the nesting algorithm. The simulation of Berkhoff et al.'s (1982) experiment showed that the model is capable of simulating surface waves and their transformation in shallow water, which involves dispersive and nonlinear effects. The simulation of experiments for solitary wave runup on a shelf with an island was used to assess the accuracy of the wetting and drying processes in the nested grid system. The last application, the simulation of the Tohoku-Oki 2011 tsunami and its effects on Crescent City Harbor, CA, demonstrated the robustness of



the two-way nesting model for the multi-scale modeling tof ransoceanic tsunamis and their coastal effects.

Future work will include the development of an interface for the GPU version of FUNWAVE-TVD and of an adaptive mesh refinement algorithm for the nesting framework.

*Code availability and data availability.* The computer code, all examples illustrated in the paper, MATLAB post-processing scripts, and data used in this research are archived at (http://doi.org/10.5281/zenodo.4735599).

*Author contributions.* The conceptualization and methodology of the study were developed by Shi and Choi. The code was implemented by Choi. Code validation was conducted by Malej. Smith, Kirby, and Grilli performed result analysis and paper revision.

*Competing interests.* The authors declare that they have no conflict of interest.

## Acknowledgments

The corresponding author, Fengyan Shi, was supported in part by an appointment to the Department of Defense (DOD) Research Participation Program administered by the Oak Ridge Institute for Science and Education (ORISE) through an interagency agreement between the U.S. Department of Energy (DOE) and the DOD. ORISE is managed by ORAU under DOE contract number DE-SC0014664. All opinions expressed in this paper are the author's and do not necessarily reflect the policies and views of DOD, DOE, or ORAU/ORISE. This work was supported by the U.S. Army Engineer Research and Development Center. Permission to publish this paper was granted by the Chief of Engineers, U.S. Army Corps of Engineers.

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
