# Peer review of "Block-structured, equal workload, multigrid nesting interface for Boussinesq wave model FUNWAVE-TVD"

_EGUsphere, 2022_

## Referee Comment (RC2)

**General Comments:**

Transoceanic tsunamis may cause serious damages to the highly populated coastal areas, such that early warnings based on the forecast of tsunami considering its coastal impact are very important. Using multi-grid nesting, this paper presents a very promising numerical tool which could simultaneously simulate the evolution of tsunami at the oceanic basin-scale with a coarse grid and the coastal inundation in the nearshore-scale with a finer grid. The four different tests demonstrate this capability impressively. The paper is well-written, and the organization of content is great. However, if the author could present more information about the background of FUNWAVE-TVD, more details about the grid/memory management, and more results about model accuracy/efficiency will improve readers' understanding about the equal workload, multigrid nesting interface. My suggestions are given as below.

**Specific Comments:**

1. Introduction (section 1) is not well-organized or lack of some information. Part of this is explained in "Technical Corrections". My suggestion is to re-organize the content following this manner: a) governing equations and explain why choose the dispersive ones. I have the same feeling as the other reviewer that the author needs to explain the need of finer grid and the relation to dispersion property more clearly; b) techniques for multi-scale tsunami modeling, including AMR and nested grids. First, introduce the AMR and why it is not ok for your governing equations; c) Second, introduce nested grids with figure 1; d) for nested grids, why two-way nesting is necessary or useful; e) summarize what has been done/ innovations (interpolator/restriction operator, data management, etc…) of this work.

2. Section 2 (FUNWAVE-TVD). First, it could be better to move lines 150-160 here and add a subsection just for the development relating to FUNWAVE. Second, use one paragraph in section 2.1 (no need to separate the content). Third, section 2.2 is over-simplified. Even though this part is not that important in this paper, enough information about the numerical schemes used by FUNWAVE-TVD is necessary. What's more, one thing should be explained is that the core program FUNWAVE-TVD only needs a grid (parent or child), an initial condition, and the boundary conditions in ghost cells to launch. This is necessary as a precondition for the multi-grid nesting interface. Last, use a figure similar to figure 3 to explain the penalization and ghost cells in section 2.3.

3. Algorithms in section 3 are not introduced clearly. First, the restriction operator should be important, but I do not know what it is. Please provide an equation. Workload balance is easy in section 3.3, but what I am interested in is the data management. Could please provide more details regarding the shared memory and grid management? Last, figure 4 could be improved with condition judgement, details of grid/variables/boundary assignment.

4. Section 4, Applications. I understand that the authors mentioned the units for figures. But it could be more straightforward to include this information along with the figure colorbar. In addition, I know that the present model has been verified with laboratory data. But it could be great to add a fine-grid simulation in section 4.1with a single grid resolution of 1.25m and do the comparison. This way, it is clearer to demonstrate the effectiveness of the present work. Last, improve the axis labels of figures 15 and 16.

5. I'm personally interested in one question that, is the code suitable for grid refinement for two separated areas?

**Technical Corrections:**

| No. | Location | Comments |
| --- | --- | --- |
| 1 | Line 5-10 | Address format is not consistent |
| 2 | Line 14 | Cite the paper for FUNWAVE-TVD as there exists several versions. |
| 3 | Line 16 | The nesting interface has more functions, like grid/memory management. |
| 4 | Line 17 | "child grids" |
| 5 | Line 19 | Remove comma in "data management, " |
| 6 | Line 21 | "to verify the nesting algorithm, to assess model accuracy…" |
| 7 | Line 22 | "modeling" to "model" |
| 8 | Line 35 | Remove "and accuracy". Strategies in workload balance, data management, and parent-child communications does not guarantee accuracy. |
| 9 | Line 31-36 | I cannot find the advantage of the new interface here. At least, it should not be a repeat of the Abstract. |
| 10 | Line 45 | "are typically based on" |
| 11 | Line 47 | Remove "or" with "," |
| 12 | Line 48 | ", or on" |
| 13 | Line 40-64 | This part has explained why using dispersive models. But, why do not choose the non-hydrostatic models? |
| 14 | Line 79 | "coarse grid to the fine grid", opposite? |
| 15 | Line 65-82 | The reason for two-way nesting is not clear. What if the feedback from fine grid to coarse grid is small? Provide some references. Somehow, this has been explained in lines 101-120. |
| 16 | Line 150-167 | The prime work should be summarized here, not the objective. |
| 17 | Line 171 | "the present study" |
| 18 | Line 186-187 | "Governing equations" is good enough. |
| 19 | Between line 189 & 190 | Explain the dimensionless parameter mu. |
| 20 | Line 191 | "equation (4)". Check the whole paper, please. |
| 21 | Line 232 | "two-way" |
| 22 | Line 230-233 | Not necessary. |
| 23 | Line 246 | operators |
| 24 | Line 247-256 | This can be explained in the end of Introduction, along with interpolator and restriction operators. |
| 25 | Equation (10) | Use symbols other than t and s as they are used for time and grid refinement ratio. |
| 26 | Line 313 | "a time step" |
| 27 | Line 315 | "program is called" |
| 28 | Figure 4 | Explain the flowchart where a condition (two directions) is checked. |

| 29 | Equation above line 331 | Please add dimension. |
|---|---|---|
| 30 | Figure 6 | Four different lines are plotted but I can only distinguish part of them. Figure 6 is used to show that the evolution of elevation at four locations are almost the same, but this can be explained in the text without an extra figure. Or, if the accuracy is important, why not adding information about the relative difference? |
| 31 | Figure 10 | Some figures use (a), but this figure uses (1)-(8) |
| 32 | Line 393-394 | If the test 4.2 is not a typical case for demonstrating the efficiency of the nested grid method, could you give us a good demonstration as the I regard the computational efficiency is very important. Also, it will be a great promotion of this work. |
| 33 | Line 512-513 | Not necessary. |
| 34 | Line 533 | Please check the full references.
Geophysical Research: Oceans, 122(12) |
| 35 | Line 538 | Geophysical Research Letters, 41 |
| 36 | Line 541 | "Tech. rep.", abbreviation? |
| 37 | Line 544 | Hawaii |
| 38 | Line 568 | 49 |
| 39 | Line 637 | Fuhrman, D. R. (space) |
| 40 | Line 657 | Space after doi: |
| 41 | Line 660 | 43 |
| 42 | Line 661-663 | Style is not consistent with Kirby et al. (1998) |
| 43 | Line 670-672 | Style is wrong |
| 44 | Line 674 | doi: (lower case and space) |
| 45 | Line 684-685 | could be a research report |

---

## Author Comment (AC1)

We would like to thank reviewer for providing useful comments/suggestions towards the improvement of our manuscript. In the following responses, the reviewer's questions are printed in black and our responses are in blue. In the revised manuscript, the changes are highlighted in red.

**Response to Reviewer 1**

The manuscript describes an extension of a popular wave model with the aim to increase applicability and computational speed through multi-grid nesting and MPI parallelization. The paper is well-written and most sections are properly explained. However, there are still a few section, where more information in necessary. This applies mostly to memory sharing, synchonizaion and several other small details, which could really help a less experienced researcher in understanding the details of this presented technique. Below are a few comments, which can help to improve the overall quality and message of this paper.

Reply: We noticed that the major concern from the reviewer was the clearness of the descriptions about the nesting algorithm and techniques associated with data structures and workflow management. In the revised version, we made a major revision in section 3 and tried to better present the detailed techniques used in the master program. We also clarified a number of confusing statements pointed out by the reviewer. According to the reviewer's suggestions, we tried to use plain language and simple examples, which may help users with less experience in computer programming to understand the model nesting procedures in a parallelized system. The following are detailed responses to the reviewer's questions.

Page 4 line 101-104 :
    "is 4 or better" meaning refinement factor >=4 or 4<= ?

Reply: In the revision, we reworded the sentence as "grid refinement ratio (. . . ) is 4 or smaller. "
Page 7 line 176-183: Any particular reason why the spherical mode solve the weakly nonlinear equations and not the fully nonlinear equations?

Reply: We think this is a very good question. The spherical model solves the weakly nonlinear Boussinesq equations derived by Kirby et al. (2013). The weakly nonlinear equations were developed because a spherical model is usually used in basin-scale applications where the nonlinearity is relatively weak. However, the recent applications in the National Tsunami Hazard Mitigation Program (NTHMP) have shown that it is more convenient to make multi-grid nesting in the single spherical mode, rather than a combination of the spherical and Cartesian modes, for simulations of transoceanic tsunamis and coastal effects in terms of map projections. We think the reviewer made a good point that the spherical mode should be further developed into a fully nonlinear model, consistent with the Cartesian mode. To address the question, we add a short statement in the future work in the conclusion section:

"As mentioned at the beginning of the paper, a combination of the weakly nonlinear spherical mode and the fully nonlinear Cartesian mode has often been used in the one-way coupling nested grid framework for transoceanic tsunami simulations. However, the nesting interface developed in the present study cannot be used for such mixed-mode applications. It is necessary to further develop the spherical mode based on the fully nonlinear Boussinesq equations. This development may be left as future work, noting that any new developments in the model will not interfere with the nesting interface developed here. Future work may also include the development of an interface for the GPU version of FUNWAVE-TVD and of an adaptive mesh refinement algorithm for the nesting framework. "

Page 10 line 225-228: Is it necessary to exchange the dispersive terms: how does it increase the model efficiency.

Reply: The strategy of directly exchanging dispersive terms can increase the model efficiency by avoiding recalculating those dispersive terms inside the ghost cells.

How does the exchange work for the tridiagonal solver (child grid)?

Reply: In a child grid, the tridiagonal solver is performed after the data exchange

in ghost cells. The tridiagonal matrix is solved for the given boundary conditions $(u_\alpha, v_\alpha)$ at the ghost cells. In the revised version, we clarified this in section 3.1.

Page 12 line 275-276: The restriction operator (The update operator) is not detailed. Which parent grid cells are actually updated? What about the ghost cells? What variables are updated in the Parent grid? Free surface/ velocity/ dispersion terms?

Reply: We added detailed description of the restriction operator.

"The restriction operator can be expressed as

$$\varphi_2 = \frac{1}{MN} \sum_{i=1}^{M} \sum_{j=1}^{N} \tilde{\varphi}_{i,j} \tag{1}$$

where $\varphi_2$ represents the averaged value passing to the parent grid, $\tilde{\varphi}_{i,j}$ is the value at $(i, j)$ in the child grid, $(M, N)$ represent the numbers of child grid cells in $(x, y)$ directions embedded in each parent grid cell.

Both the interpolator and the restriction operator are performed to $\eta, u_\alpha, v_\alpha, U$ and $V$, which pass the values back and forth between the child domain $\Omega_1$ and the parent $\Omega_0$ in the two-way nesting process. It should be pointed out that $U$ and $V$ are not necessarily included in the interpolation/restriction processes because they can be calculated based on $\eta, u_\alpha$, and $v_\alpha$. However, our tests show that directly using the passed $U$ and $V$ can make the model more efficient and does not affect the results much. "

Workload balance and data management
Page 13 line 293-295: This statement is not clear. What terms are pre-computed before the model run?

Reply: We rewrote this statement, providing more details on how to get the parent-child proximity:

"For efficient communication between the parent and child grids in the MPI-based parallel communication system, parent-child spatial proximity is created at the beginning of the model run. The parent-child proximity is associated with the

parent processor IDs and the child processor IDs at the nesting boundaries in the image distribution, as shown in Fig. 3. The parent-child proximity records the spatial relation of processor IDs between the parent and child grids and is saved in a parameter array. Therefore, the communication between the parent and child grids can be carried out straightforwardly using MPI_SEND and MPI_IRECV (MPI library) with the existing parameter array whenever a communication is needed. In this example, along the west boundary of the child grid, the child processors with ID=1, 2, and 3 communicate directly with parent processors with ID = 4 and 5. In particular, the variables in the ghost cells located in child processor ID=1 are evaluated by the interpolator in the parent processor ID=4, and so on."

The child-parent proximity (if proximity means boundaries) changes at each Parent time step – Same for the restriction process.

Reply: In the revision, we used child-parent spatial-proximity to avoid confusing. It is saved in a parameter array which won't change with time.

Personally I think the implementation is not very detailed considering that it's the main contribution from the paper. I don't understand how the communication between the child and the parent grid is straightforward...

Reply: We hope the rewriting based on the last question (parent-child proximity) can help to understand better the communication algorithm.

When the authors talk about shared array allocation, do all processors have a copy of all the model variables (Parents and child grids) + the boundary condition of child grid ? How does the synchronization work?

Reply: The shared array allocation here means that an allocated array (a variable) can be shared by all grids (parent and child), and thus that no additional allocation is needed for a particular grid. To make the definition more clearly, we use the example of the rectangular-shaped hump presented in 4.1.

" A strategy of shared array allocation is used, whereby the arrays are allocated

with the maximum dimension of all grids at the initialization stage, and grids at all levels share the same memory allocations. For example, in the case presented later in section 4.1, the nesting system has four grid levels, the dimensions of which are (48, 48), (73, 73), (92, 92), and (91, 91) for grids 1-4, respectively. A two-dimensional array for $\eta$ will be allocated in $\eta(92,92)$, the maximum dimension among the four grids. The four grids share the same array $\eta(92,92)$, while the grids with a dimension smaller than (92,92) only use part of allocation, i.e., (48,48) for grid 1, (73,73) for grid 2, and (91, 91) for grid 4. There is no additional array allocation needed for a specific child grid with such a shared allocation strategy. It can apparently save a significant amount of computer memory and does not create an extra burden for a large number of nested grids. "

The nesting processes are actually sequential. The synchronization is performed at each grid level as in a single grid FUNWAVE model. No processor is idle because of the equal computational load.

We are sorry that the description in the last manuscript is not clear. In the revision, we used the example above to explain this concept.

How is the MPI implementation optimized for nested grid?

Reply: As mentioned in the last question, the calculations from one grid to another are sequential. The grid partition is performed at each grid level with the equal division (not exactly due to non-divisible grid numbers). There is no additional optimization for MPI partitions for nested grids.

In section 3.4, we added

"In summary, the hierarchical-type grid refinement processes are sequential and the synchronization is conducted at each grid level. Therefore, there is no additional optimization for MPI partitions needed for the nesting framework. "

How do the authors synchronize the parent and child grid after each parent time step? Do the authors use a fixed time step for the Child grid? If they use the CFL condition for the parent and the child solutions, does this involve that some type of synchronization is required before the update step.

Reply: We used CFL-criterion. $\Delta t$ for the first grid level is determined by the CFL-criterion and is time varying. Because the subgrid ratio is an integer number, the time steps for child grid levels are reduced correspondingly based on the criterion. In the revised version, we added the following description.

"Based on the assumption that the grid refinement factor equals the time refinement factor, and the linear relation between time step and grid spacing holds in the CFL criterion, the child time step between two parent time levels is constant and can be written as

$$\Delta t_{\text{child}} = \Delta t_{\text{parent}}/s \tag{2}$$

where, $\Delta t_{\text{parent}}$ and $\Delta t_{\text{child}}$ denote the parent time step and child time step, respectively. "

Because the hierarchical-type grid refinement processes are sequential there is no synchronization involved in the nesting process.

Application

4.1 Evolution of an initial rectangular-shaped hump

Symmetry test is okay

Figure 7 : maybe include a diagonal transect and plot (a) (b) and (c) in the same figure. For better comparison.

Reply: We updated Figure 7 as suggested and also added a subplot showing a detailed comparison between different grid configurations along a transect.

Page18 line 349-352: The whole solution depends on grid resolution not only dispersive effects.

Reply: We agree with the reviewer that the solution is not only dependent on dispersive effects but also the grid resolution which can resolve the dispersive undulations.

ADDITIONAL CORRECTIONS

1) We re-drawed Figures 3, 8 and 11 to make consistency with the text.

2) We found an error in the number of ghost cells used in the nesting boundary. It should be 4, not 3. We corrected Figure 3, which is related to the issue and added the text (in section 3.1)

"Four ghost cells are used along $\Gamma$. Here, we want to mention that the number of ghost cells in the MPI-based parallelization is three, as required by the higher-order numerical schemes used in the model. The nesting scheme needs one more cell for the requirement because of re-calculating rather than passing values of the ghost cells. "

3) Some typos were corrected.

We want to thank the reviewer again for providing such a detailed and constructive review.

---

## Author Comment (AC2)

We would like to thank the reviewer for detailed suggestions and comments which are constructive and helpful for improving the paper quality. In the following responses, the reviewer's questions are printed in black and our responses are in blue. In the revised manuscript, the changes are highlighted in red.

**Response to Reviewer 2**

**General Comments:**
Transoceanic tsunamis may cause serious damages to the highly populated coastal areas, such that early warnings based on the forecast of tsunami considering its coastal impact are very important. Using multi-grid nesting, this paper presents a very promising numerical tool which could simultaneously simulate the evolution of tsunami at the oceanic basin-scale with a coarse grid and the coastal inundation in the nearshore-scale with a finer grid. The four different tests demonstrate this capability impressively. The paper is well-written, and the organization of content is great. However, if the author could present more information about the background of FUNWAVE-TVD, more details about the grid/memory management, and more results about model accuracy/efficiency will improve readers' understanding about the equal workload, multigrid nesting interface. My suggestions are given as below.

Reply: As we can understood from the general comments and the following specific comments, the reviewer suggested clarifying and providing more details in some critical techniques used in the development. We think that this suggestion is in line with the general comment from Reviewer 1. In this revision, we added some explanations and clarifications related to the techniques mentioned by both of the reviewers, such as memory management and equal workload algorithm. As suggested by the reviewer, we carried out an extra test in the case of the rectangular hump using a high resolution, single grid configuration. We found that the result is interesting and good to be added in the paper. We also made an effort on correcting figures and texts following the detailed comments from the reviewer.

**Specific Comments:**

1. Introduction (section 1) is not well-organized or lack of some information. Part of this is explained in "Technical Corrections". My suggestion is to re-organize the content following this manner: a) governing equations and explain why choose the dispersive ones. I have the same feeling as the other reviewer that the author needs to explain the need of finer grid and the relation to dispersion property more clearly; b) techniques for multi-scale tsunami modeling, including AMR and nested grids. First, introduce the AMR and why it is not ok for your governing equations; c) Second, introduce nested grids with figure 1; d) for nested grids, why two-way nesting is necessary or useful; e) summarize what has been done/ innovations (interpolator/restriction operator, data management, etc...) of this work.

Reply: We appreciate the 'Technical Corrections' provided by the reviewer. Our corrections are attached at the end of the letter. Following the reviewer's suggestion, we reorganized the introduction section in order of dispersion importance –> nesting techniques –> work in this study. We emphasized that, 1) the Boussinesq model is more appropriate model relative to other types of dispersive wave model in tsunami simulations, 2) why two-way nesting is necessary, and 3) an equal-load scheme in workload balance and a strategy of shared array allocation for data management. Since some AMR techniques are closely related to the two-way nesting method, we decided to introduce the nesting method prior to the AMR methods. FUNWAVE equations are also good for AMR. We mentioned that the development of the AMR algorithm is left for future work (see conclusion section).

2. Section 2 (FUNWAVE-TVD). First, it could be better to move lines 150-160 here and add a subsection just for the development relating to FUNWAVE. Second, use one paragraph in section 2.1 (no need to separate the content). Third, section 2.2 is over-simplified. Even though this part is not that important in this paper, enough information about the numerical schemes used by FUNWAVE-TVD is necessary. What's more, one thing should be explained is that the core program FUNWAVE-TVD only needs a grid (parent or child), an initial condition, and the boundary conditions in ghost cells to launch. This is necessary as a precondition

for the multi-grid nesting interface. Last, use a figure similar to figure 3 to explain the penalization and ghost cells in section 2.3.

Reply: We agree with the reviewer that section 2 should be better organized. We combined the mentioned paragraph and the first paragraph of section 2 into a new subsection 2.1. In the numerical scheme subsection, we mentioned that several optional schemes with different orders of accuracy were implemented. For this reason, we only briefly present the basic methods related to the nesting procedure in this study. Readers interested in detailed numerical schemes are referred to Shi et al. (2012) and Choi et al. (2018). In the last paragraph of the numerical scheme subsection, we pointed out that the multi-grid interface is developed separately from the core program, and each grid in the nesting system runs the same core program for given initial and boundary conditions.

Because the parallelization scheme used in FUNWAVE-TVD strictly follows the MPI algorithm, especially the boundary data exchange method, we decided not to provide an extra figure to avoid duplication of this popular method already described in other documentations. We did provide some text to explain the data exchange through ghost cells. We hope the revision in this fashion is satisfactory.

3. Algorithms in section 3 are not introduced clearly. First, the restriction operator should be important, but I do not know what it is. Please provide an equation. Workload balance is easy in section 3.3, but what I am interested in is the data management. Could please provide more details regarding the shared memory and grid management? Last, figure 4 could be improved with condition judgement, details of grid/variables/boundary assignment.

Reply: In this revision, we provided a formulation for the restriction operator. In section 3.3, we gave more detailed descriptions about the shared memory/data management methods. Because this suggestion is closely consistent with reviewer 1's suggestion, we do not repeat the detailed revision here.

We modified figure 4 based on the suggestion.

4. Section 4, Applications. I understand that the authors mentioned the units

for figures. But it could be more straightforward to include this information along with the figure colorbar. In addition, I know that the present model has been verified with laboratory data. But it could be great to add a fine-grid simulation in section 4.1with a single grid resolution of 1.25m and do the comparison. This way, it is clearer to demonstrate the effectiveness of the present work. Last, improve the axis labels of figures 15 and 16.

Reply: Following the suggestions, we redrew Figures 5, 7, 9, 15, and 16 with colorbar added and better look for axis labels. We carried out an extra test on the rectangular hump case using a single grid with a resolution of 1.25m. It turned out to be very interesting test showing a strong dispersion effect. The dispersive short waves appearing in the finest nested grid (1.25 m resolution) are consistent with the dispersive wave pattern from the single fine grid simulation. In the revision, we plotted the single grid result in a subplot of the original Figure 7 (Figure 6 in the revised version) and added an additional figure (Figure 7 in the revised version) showing detailed comparisons of surface elevation along a section. The original Figure 6 was removed according to question 30 in Technical Correction.

5. I'm personally interested in one question that, is the code suitable for grid refinement for two separated areas?

Reply: Thanks for your interest. The current version of the code cannot deal with two or more separate areas. However, it is feasible to add this feature in the MASTER program. This work should be included in the development of the adaptive mesh refinement algorithm as mentioned in the future work.

**Technical Correction:** (NOTE: minor corrections, such as typos, are not marked in red in the main text)

| No. | Location | Comments | Responses |
|---|---|---|---|
| 1 | Line 5-10 | Address format is not consistent | Corrected |
| 2 | Line 14 | Cite the paper for FUNWAVE-TVD as there exists several versions. | Cited in main text |
| 3 | Line 16 | The nesting interface has more functions, like grid/memory management. | Mentioned in later sentence |
| 4 | Line 17 | "child grids" | We used singular because we mean between the two generations, a child grid could be a parent grid of next generation. |
| 5 | Line 19 | Remove comma in "data management, " | Corrected |
| 6 | Line 21 | "to verify the nesting algorithm, to assess model accuracy…" | Corrected |
| 7 | Line 22 | "modeling" to "model" | Corrected |
| 8 | Line 35 | Remove "and accuracy". Strategies in workload balance, data management, and parent-child communications does not guarantee accuracy. | Corrected |
| 9 | Line 31-36 | I cannot find the advantage of the new interface here. At least, it should not be a repeat of the Abstract. | We reworded the sentence. In fact, the language used in the abstract is close to the plain language. |
| 10 | Line 45 | "are typically based on" | Corrected |
| 11 | Line 47 | Remove "or" with "," | Corrected |
| 12 | Line 48 | ", or on" | Corrected |
| 13 | Line 40 - 64 | This part has explained why using dispersive models. But, why do not choose the non-hydrostatic models? | We clarified this by adding a sentence at the end of the paragraph. |
| 14 | Line 79 | "coarse grid to the fine grid", opposite? | You are right. Corrected |
| 15 | Line 65 – 82 | The reason for two-way nesting is not clear. What if the feedback from fine grid to coarse grid is small? Provide some references. Somehow, this has been explained in lines 101-120. | We added the example of one-way nesting effect on edge wave motions appearing in the case of Crescent City Harbor (Tehranirad et al. 2020) |
| 16 | Line 150 - 167 | The prime work should be summarized here, not the objective. | Following the earlier suggestion, we moved this paragraph to section 2.1 and rewrote the summary at the end of this section. |
| 17 | Line 171 | "the present study" | Corrected |
| 18 | Line 186 - 187 | "Governing equations" is good enough. | Corrected |
| 19 | Between line 189 & 190 | Explain the dimensionless parameter mu. | Added the definition. |
| 20 | Line 191 | "equation (4)". Check the whole paper, please. | We checked consistency over the entire text. We use parentheses only for equations |
| 21 | Line 232 | "two-way" | Corrected |
| 22 | Line 230 - 233 | Not necessary. | Removed |
| 23 | Line 246 | operators | We use two terms, "interpolator" |

| | | | and "restriction operator". To avoid confusing,
we changed interpolators in the text to interpolator to keep consistency |
|---|---|---|---|
| 24 | Line 247 - 256 | This can be explained in the end of Introduction, along with interpolator and restriction operators. | We added in the introduction. Detailed formulas are given here. |
| 25 | Equation (10) | Use symbols other than t and s as they are used for time and grid refinement ratio. | We corrected to a and b |
| 26 | Line 313 | "a time step" | Corrected |
| 27 | Line 315 | "program is called" | Corrected |
| 28 | Figure 4 | Explain the flowchart where a condition (two directions) is checked. | We added conditions in the flowchart and defined 'time' and 'grid level' in the figure caption. |
| 29 | Equation above line 331 | Please add dimension. | Added |
| 30 | Figure 6 | Four different lines are plotted but I can only distinguish part of them. Figure 6 is used to show that the evolution of elevation at four locations are almost the same, but this can be explained in the text without an extra figure. Or, if the accuracy is important, why not adding information about the relative difference? | Removed the figure. |
| 31 | Figure 10 | Some figures use (a), but this figure uses (1)-(8) | We re-plotted several figures with a consistent (a) type |
| 32 | Line 393-394 | If the test 4.2 is not a typical case for demonstrating the efficiency of the nested grid method, could you give us a good demonstration as the I regard the computational efficiency is very important. Also, it will be a great promotion of this work. | In the text, we pointed out that the Tohoku-Oki case is a typical case for demonstrating the efficiency of the present grid nesting framework. We emphasized its automatic nesting procedure versus the labor-intensive pre-and post-processing procedures in the old nesting method. |
| 33 | Line 512 - 513 | Not necessary. | We rewrote the future work. We mentioned GPU because it may be our next major task supported by the funding agency. |
| 34 | Line 533 | Please check the full references.
Geophysical Research: Oceans, 122(12) | Corrected |
| 35 | Line 538 | Geophysical Research Letters, 41 | Corrected |
| 36 | Line 541 | "Tech. rep.", abbreviation? | Corrected |
| 37 | Line 544 | Hawaii | Corrected |
| 38 | Line 568 | 49 | The issue is `49-50' |
| 39 | Line 637 | Fuhrman, D. R. (space) | Corrected |
| 40 | Line 657 | Space after doi: | Corrected |
| 41 | Line 660 | 43 | Issue is `43-44' |
| 42 | Line 661 – 663 | Style is not consistent with Kirby et al. (1998) | Corrected |
| 43 | Line 670 - 672 | Style is wrong | Corrected |
| 44 | Line 674 | doi: (lower case and space) | Corrected |

| 45 | Line 684 - 685 | could be a research report | Corrected |
|----|----------------|----------------------------|-----------|

**Again, we would like to express our appreciation for the reviewer who provided such detailed comments/suggestions.**